# Prevalence of Mental Health Problems among Patients Treated by Emergency Medical Teams: Findings from J-SPEED Data Regarding the West Japan Heavy Rain 2018

**DOI:** 10.3390/ijerph191811454

**Published:** 2022-09-12

**Authors:** Yui Yumiya, Odgerel Chimed-Ochir, Akihiro Taji, Eisaku Kishita, Kouki Akahoshi, Hisayoshi Kondo, Akinori Wakai, Kayoko Chishima, Yoshiki Toyokuni, Yuichi Koido, Hirokazu Tachikawa, Sho Takahashi, Sayaka Gomei, Yuzuru Kawashima, Tatsuhiko Kubo

**Affiliations:** 1Department of Public Health and Health Policy, Graduate School of Biomedical and Health Sciences, Hiroshima University, Hiroshima 734-8553, Japan; 2Hiroshima Prefectural Government Health and Welfare Affairs Bureau, Hiroshima 730-8511, Japan; 3National Hospital Organization Headquarters DMAT Secretariat MHLW Japan, Tokyo 190-8579, Japan; 4Department of Disaster and Community Psychiatry, Division of Clinical Medicine, Faculty of Medicine, University of Tsukuba, Tsukuba 305-8575, Japan; 5Department of Emergency and Critical Care Medicine, Dokkyo Medical University Saitama Medical Center, Saitama 343-8555, Japan; 6DPAT Secretariat, Tokyo 108-8554, Japan

**Keywords:** Emergency Medical Team, Emergency Medical Team minimum data set, J-SPEED, natural disasters, epidemiology, sentinel surveillance, disaster psychiatry, mental health

## Abstract

It is crucial to provide mental health care following a disaster because the victims tend to experience symptoms such as anxiety and insomnia during the acute phase. However, little research on mental health during the acute phase has been conducted, and reported only in terms of the temporal transition of the number of consultations and symptoms. Thus, the aim of the study was to examine how mental health care needs are accounted for in the overall picture of disaster relief and how they change over time. Using data from the Japanese version of Surveillance in Post-Extreme Emergencies and Disasters (J-SPEED), we assessed the mental health of injured and ill patients to whom Emergency Medical Teams (EMTs) were providing care during the acute period of a disaster. Approximately 10% of all medical consultations were for mental health issues, 83% of which took place within the first 2 weeks after the disaster. The findings showed that, from the start of the response period to the 19th response day, the daily proportion of mental health problems declined substantially, and then gradually increased. Such a V-shaped pattern might be helpful for identifying phase changes and supporting the development of EMT exit strategies.

## 1. Introduction

The provision of mental health care from the acute phase after a disaster is critical. According to a previous epidemiological study, disaster victims tend to experience mental health symptoms such as anxiety, sleep problems, and mood and affect during this period [1]. Although mental support is required in the acute phase, a disaster is a situation that exceeds the capacity of a community to respond, and therefore requires external support. Thus, Emergency Medical Teams (EMTs), including Disaster Medical Assistance Teams (DMATs), the Japan Medical Association Team (JMAT), and the Japan Red Cross Medical Assistance Team (JRC), are dispatched to affected areas to provide not only physical, but also psychological care, such as psychological first aid (PFA), during disasters [2]. Furthermore, in Japan, Disaster Psychiatric Assistance Teams (DPATs) are dispatched to provide mental health medical assistance in addition to a well-organized group that specializes in psychiatric medical treatment during the acute phase. Worldwide, specialized mental health support from the acute phase is rare, with a somewhat medium-to long-term focus in the form of mental health and psychosocial support (MHPSS) being more common [3]. Although mental health support from the acute phase after a disaster is considered important, opinions differ on the timing of support, with disagreements regarding when it is at its maximum and for how long it is needed. In addition, regarding post-disaster psychology, victims are considered to be anxious at first, followed by a honeymoon period during which their psychology temporarily improves and then individually worsens [4]. Yet, to our knowledge, little research has been conducted on such psychological changes during a disaster [5].

Regarding epidemiological studies on mental health in the aftermath of a disaster, numerous studies regarding the medium-and long-term effects of disasters on mental health have been reported [6]. However, studies describing mental health in the acute stage of a disaster remain extremely limited, and the provision of mental support by DPATs has been reported only in terms of the temporal transition of the number of consultations and symptoms [1,7]. Therefore, to allocate limited support resources appropriately, it is necessary to standardize the proportion and temporal transition of mental health needs in the overall picture of disaster support.

In Japan, data on activities during disasters are collected through the Japanese version of Surveillance in Post-Extreme Emergencies and Disasters (J-SPEED) [8], which is similar to the World Health Organization (WHO) EMT Minimum Data Set (MDS), a strategy for standardized medical data collection during and after a disaster [9]. Therefore, in this study, we used data from J-SPEED on the West Japan Heavy Rain 2018 [10], which included about 15,300 evacuees, 212 deaths, 342 injured, and 38,820 houses damaged among Okayama, Hiroshima, and Ehime Prefectures [11]. The aim of this study is to examine how mental health needs are accounted for in the overall picture of disaster relief and how they change over time.

## 2. Materials and Methods

### 2.1. Study Design

This study involves a descriptive analysis of daily reports collected by EMTs during the West Japan Heavy Rain 2018 that occurred from 8 July through 11 September 2018.

### 2.2. Data Collection

The numbers and types of health problems treated by EMTs in accordance with the J-SPEED (Ver 1.0) form were reported on a daily basis by EMTs to the EMT Coordination Cell. From the total number of health problems, mental health problems were extracted and analyzed in the present study. Disaster stress-related symptoms included insomnia, headache, dizziness, loss of appetite, gastric pain, and constipation, and mental care needs included suicide attempts, problem behaviors and attitudes, and restlessness. A detailed description of the J-SPEED form and data collection has been published [10].

### 2.3. Data Analysis

First, we descriptively analyzed the mental health problems by age group (<1, 1–8, 9–74, and ≥75 years) and day of response. Second, we used a joinpoint regression model to study trends in the number and proportion of mental health problems from among the total number of health events over the response period. Although mental health problems were reported for 41 days between 8 July to 11 September 2018, the joinpoint regression analysis did not include the period after day 27 because fewer than 10 daily reported total consultations were typically found, and calculating the percentage from such a small number does not add useful information.

To minimize the effect of disparities in the number of EMTs during the response period, 7-day moving averages were calculated for the number of mental health consultations and the proportion of mental health problems to the total number of consultations. The 7-day moving average of mental health consultations was divided by that of total consultations to calculate the proportion of mental health problems to the total number of consultations.

The model first identified the time points that showed a significant shift in trends and estimated the daily percentage change (DPC), which indicates the rate of average change between two joinpoints.

To estimate the DPC, the following regression model was used:log(*Rt*) = b0 + b1*t*,(1)
where log(*Rt*) is the natural log of the rate in day *t*.
(2)$$\mathrm{The}\;\mathrm{DPC}\;\mathrm{from}\;\mathrm{day}\;t\;\mathrm{to}\;\mathrm{day}\;t+1=\frac{R_{t+1}-R_{t}}{R_{t}}\times \;100=\frac{\left\{e^{b_{0}+b_{1}\left(t+1\right)-}e^{b_{0}+b_{1}\left(t\right)}\right\}}{e^{b_{0}+b_{1}\left(t\right)}}\;\times \;100=\left(e^{b_{1}}-1\right)\;\times \;100$$

The DPC over any fixed interval is calculated using a weighted average of the slope coefficients of the underlying joinpoint regression line, with the weights equal to the length of each segment over the interval. The final step of the calculation transforms the weighted average of slope coefficients to an annual percent change. If we denote *b* as the slope coefficient for each segment in the desired range of days, and *w* as the length of each segment in the range of days, then
(3)$$\mathrm{ADPC}=\left\{Exp\left(\frac{\sum w_{i}b_{i}}{\sum w_{i}}\right)-1\right\}\;\times \;100$$

The Joinpoint Regression Program does not provide models that best fit the data, but rather, models that best summarize the behavior or data trend [12]. Joinpoint tests of significance use a Monte Carlo permutation method to determine statistically significant changes across successive calendar periods, and the average daily percent change (ADPC) was estimated utilizing generalized linear models assuming a Poisson distribution [13]. A more detailed mathematical explanation can be found elsewhere [13].

Microsoft Excel (Microsoft Corp.; Redmond, WA, USA) and STATA v15.1 (STATA Corp; College Station, TX, USA) were used for the analysis.

## 3. Results

The number of consultations for mental health problems during the West Japan Heavy Rain 2018 is shown in Table 1. A total of 372 (10.3%) mental health consultations were recorded out of 3617 total consultations. Among these, 205 (9.9%) were for adults aged 9–74 years, and 161 (11.5%) were for people aged ≥75 years.

Figure 1 shows the total number of mental health consultations by response day and age group. Of the 372 consultations recorded, the majority (*n* = 309; 83.1%) occurred within the first 2 weeks. During the first 2 weeks, the proportion of patients aged 9–74 years was larger than that of patients aged ≥75 years; however, patients aged ≥75 years were more likely to visit later.

Figure 2 shows the results of a joinpoint regression analysis of the 7-day moving average of daily reported mental health consultations for all ages, 9–74 years, and ≥75 years. The 7-day moving average is placed at the last point of the 7-day interval. For example, the 7-day moving average for days 1 through 7 is placed on day 7. In other words, the joinpoint regression analysis was carried out for the period day 7 through day 26, although it covered the daily reported mental health consultations from day 1 to day 26. The daily number of mental health consultations significantly increased for all ages (ADPC = 43.26%; *p* < 0.0001) and those aged ≥75 years (ADPC = 59.56%; *p* < 0.0001). However, at 2 weeks after the day of response, the daily number of mental health consultations significantly decreased for all ages (ADPC = 23.66%; *p* < 0.0001), those aged 9–74 years (ADPC = 27.71%; *p* < 0.0001), and those aged ≥75 years (ADPC = 21.31%; *p* < 0.0001).

Figure 3 shows trends in the 7-day moving average regarding the proportion of mental health consultations to all consultations. The daily percentage of mental health consultations significantly decreased from the beginning of the response period until response day 19 for all ages (ADPC = 6.75%; *p* < 0.0001), those aged 9–74 years (ADPC = 9.75%; *p* < 0.0001), and those aged ≥65 years (ADPC = 3.89%; *p* < 0.0001), and then increased by around 30% for all groups.

## 4. Discussion

We characterized the mental health status of injured and ill victims to whom EMTs responded in the acute phase of the 2018 West Japan Heavy Rain using J-SPEED data. Mental health problems represented about 10% of all health consultations, with around 83% occurring within the first 2 weeks. Our main finding was that the daily percentage of mental health problems significantly decreased from the beginning of the response period to the 19th response day, and then gradually increased. Such a V-shaped pattern is similar to the common phase of a disaster in terms of community and individual responses [4]. Our findings also revealed a high percentage of consultations at the beginning of the response period (from day 1 to day 7). A previous report on four disasters indicated that the number of consultations peaked in the super-acute phase (ranging from 2 to 7 days) [7]. Although the number and ratio of consultations depend on the scale of the disaster, the number of dispatched EMTs, and the day when they arrive, among other factors, it can be said that there is a high mental care need during the first few weeks after a disaster. This result might be the result of a direct stress response to the disaster. Survivors may be so shocked by the event that they become psychologically paralyzed, as in cases where their own houses are damaged or destroyed by floods or landslides caused by heavy rain [14].

The following reasons may explain why the percentage of consultations decreased from the beginning of the response period to response day 19: (1) withdrawal from acute shock, (2) a sense of security and solidarity due to the arrival of relief teams, including EMTs and volunteers, and (3) psychological resilience. Disaster victims who were temporarily evacuated after the heavy rain disaster may have completed the cleanup of their houses and returned to their original lives to regain their composure, while those in shelters may have felt more secure under the care of DMAT, DPAT, and other EMTs [15,16]. In the early phase of a disaster, survivors in the community typically demonstrate altruistic behavior by rescuing, housing, feeding, and assisting their fellow humans, depending on the severity of the situation, the length of exposure, and the availability of relief resources from various organizations [4,15].

Regarding the reason for the increase in the mental health consultation rate after day 20, those who had to stay behind (those who could not immediately secure a home to return to) might feel left behind, isolated, and helpless because of fewer number of people in evacuation shelters (those who are recovering well) and the withdrawal of EMTs [15,16]. Furthermore, victims who are working hard to rebuild their own lives can easily become mentally and physically ill because of overwork and excessive stress. The previous literature review reported that secondary stressors, which arise as a result of, or are related to initial stresses, frequently appear when responding to the phases of cleanup, rehabilitation, and rebuilding following flooding [17]. In addition, the previous study revealed that residents with low psychological resilience were more likely to recover slower than were those with high psychological resilience [16]. Therefore, it is important to follow a recovery approach based on the extent of damage, phased time periods, and psychological resilience.

Taking our results together, the number of consultations requiring mental health support was concentrated within the first few weeks. Therefore, a sufficient number of EMTs who can provide MHPSS, such as DPATs, needs to be ensured. Thereafter, although a significant decrease in the number of consultations requiring mental health support was seen, the percentage of those requiring mental health care remained high. Therefore, EMTs should provide ongoing mental health care to victims so that those who need mental health care are not left behind.

The results of the present study indicate the importance of providing disaster victims with mental health services from the soon after the sudden onset of disaster. Early interventions in mental health care could prevent further mental deterioration later in life. This finding supports the technical standard [18] that all EMTs should be trained in mental health care, such as PFA, which is not a treatment, but rather a set of concepts and behaviors that can be executed by anyone [2]. Therefore, it is desirable that EMTs screen for psychiatric symptoms, identify high-risk patients through primary care, and refer them to mental health professionals [19], especially when they consult with victims who require not only physical, but also mental care. Furthermore, the DPAT system in Japan has a unique feature in that standardized teams, led by a psychiatrist and trained in mental health care, carry out planned mental health care activities beginning in the ultra-acute phase after a disaster [3]. Therefore, coordination and cooperation among all EMTs and DPATs is important to support disaster victims with possible mental issues.

Until now, phase changes have been judged mainly by the number of patients treated [7]. In the future, in addition to the number of patients treated, the percentage of patients with mental health problems may be used as an indicator to detect phase changes. In the case of an exit strategy, it is important to ensure a continuous service provision system for patients [8]. Data showing the high rate of patients with mental health issues at the time of exit may help stakeholders gain a better understanding of the importance of appropriate handovers of mental health issues to local mental health facilities. This could facilitate handover from EMTs to local facilities.

This study has some limitations. First, it is more difficult for EMTs to detect mental health issues than physical illnesses and, as a result, mental health issues are considered to be under-detected. This may be especially true in the hectic acute phase of a disaster. Although the high rate of mental health problems in the first week is rather underestimated, the conclusion that the ratio of mental health issues will remain in a V-shaped pattern is unlikely to change. Second, in this study, we did not perform a joinpoint regression model for children because of the low number of mental health problems detected among children. However, underreporting is likely to be particularly pronounced in children who are unable to communicate their symptoms. Therefore, mental health care for children is particularly important [6,20]. Further evaluations of mental health in children need to be performed based on data collected by more specialized teams with psychological experts, such as DPATs in case of Japan [20]. Moreover, the definition of mental health outcomes remains unclear, and the analysis of disaster stress-related symptoms together with urgent mental care needs may differ from the clinical picture. However, to obtain an overall picture of mental health during disasters, in the present study, two items were used for analysis. Further research using detailed items on mental health will be needed.

For this study, we could only acquire a paucity of prior relevant studies and a limited amount of data about disaster psychiatry. We explored comparative and epidemiological studies on the acute phase of a disaster in the literature, but could only find two references [1,7]. Furthermore, this study used data from only one disaster; therefore, it is challenging to generalize and compare our findings with those from other studies. However, we believe this work has unique strengths. Further case studies are needed to generalize the results to larger populations and contribute to the development of practical guidelines in the future.

## 5. Conclusions

The daily percentage of mental health problems among total health consultations showed a V-shaped pattern from the beginning of the disaster response period. This trend might be a useful indicator of phase changes and thereby support the establishment of EMT exit strategies. Although our findings rely on the results of one case study, it is expected that more data will be obtained in future studies to improve the consistency of the study results.

## Figures and Tables

**Figure 1 ijerph-19-11454-f001:**
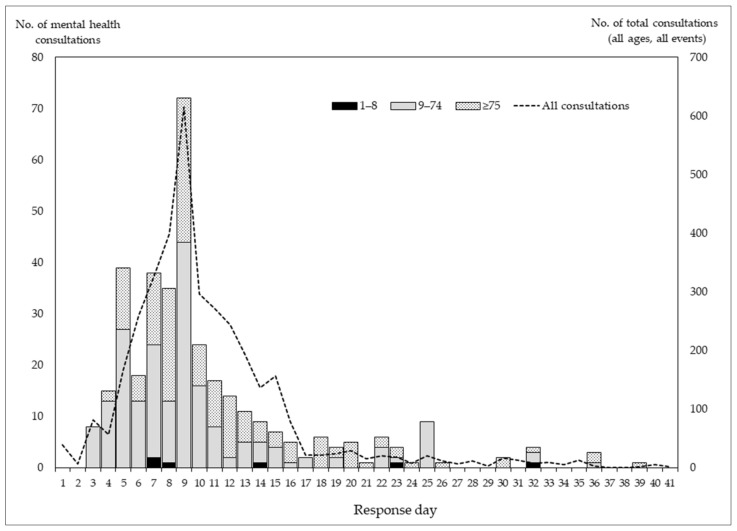
Number of mental health consultation by day of response and age group.

**Figure 2 ijerph-19-11454-f002:**
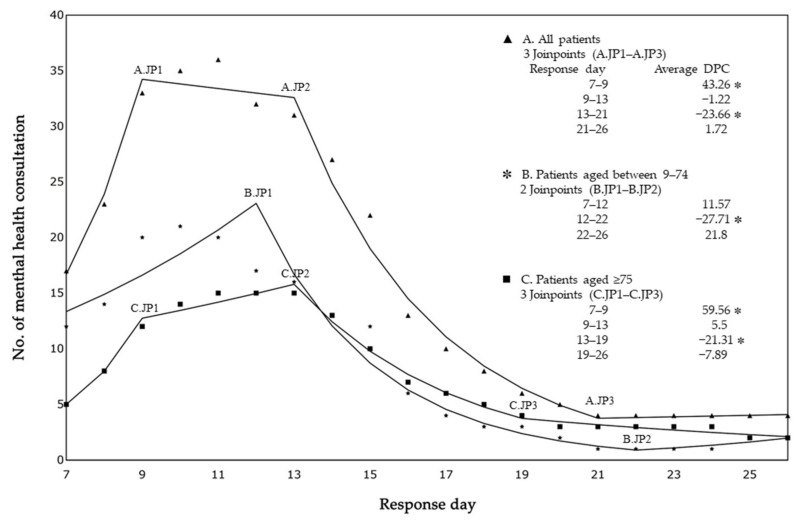
Trends in the number of mental health consultations. JP, joinpoint; DPC, daily percentage change. * Joinpoint regression analysis; *p* < 0.05.

**Figure 3 ijerph-19-11454-f003:**
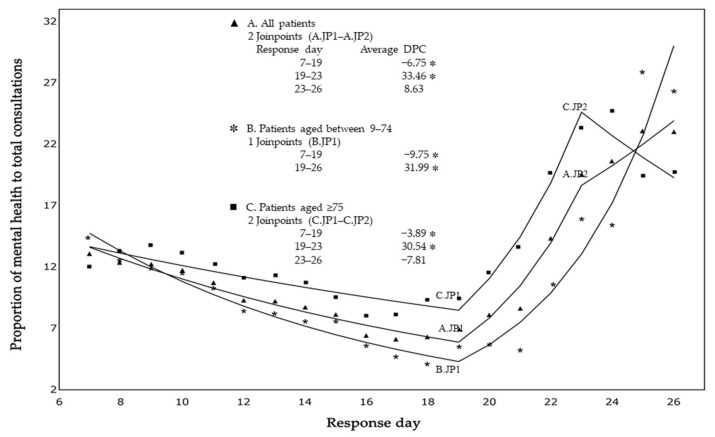
Trends in the proportion of mental health to total consultations. JP, joinpoint; DPC, daily percentage change. * Joinpoint regression analysis; *p* < 0.0001.

**Table 1 ijerph-19-11454-t001:** Summary of total and mental health consultations during the West Japan Heavy Rain 2018.

Age (years)	No. ^1^ of Total Consultations	No. ^1^ of MH ^2^ Consultations	% of MH ^2^ to Total Consultations
<1	17	0	0
1–8	139	6	4.3
9–74	2062	205	9.9
≥75	1399	161	11.5
Total	3617	372	10.3

^1^ No., Number ^2^ MH, mental health.

## Data Availability

Restrictions apply to the availability of these data. Data was obtained from the J-SPEED Research Group for research purpose and are available with the permission of the group.

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
