# Peer review of "Prevalence of Mental Health Problems among Patients Treated by Emergency Medical Teams: Findings from J-SPEED Data Regarding the West Japan Heavy Rain 2018"

_ijerph, 2022, doi:10.3390/ijerph191811454_

Round 1
Reviewer 1 Report
Dear authors, thank you for presenting this important study developed from your team.
I have some suggestions to improve your manuscript:
1) In the introduction, consider incorporate the aims of your study;
2) in the results, consider put the description of figures after the figure presentation. It is more friendly and comprehensive for the reader;
3) Maybe in the conclusions you can incorporate, although the limitations presented in the end of the results, the contributions for research, clinical practice and for society (considering the improvement of the quality of the assistance to these kinds of citizens after a disaster period.
4) There is any explicit reference to ethical procedures and authorizations. Can you incorporate these information?
Reviewer 2 Report
Material and Methods are well described, but the authors could explain the statistical method used also under the Figures. I mean just, to indicate the method used.
I suggest increasing the bibliography, especially in the discussion, which should be better argued.
Reviewer 3 Report
Dear Authors,
Many thanks for the opportunity to read your paper: it is well done and very interesting
Introduction, in a simple way, presents the topic
methods are correct and well reported
Statistics: those who have no high level of medical statistics find some difficulties: please improve the text and make it easier to understand.
Results are precise with nice graph plots
Discussion:
Do you think that the absence of a shared decision-making process could have a role in all mental diseases? Could this problem have made worse the situation ? ( doi: 10.1111/jep.13300. Epub 2019 Oct 29. PMID: 31661726) (I'm not the author)
Conclusions are adequate
